# Phonological learning and encoding with small RNNs

## 1 Introduction

With a vast expansion in available computational power, as well as the (re)discovery of some architectural and mathematical tricks, *connectionist, viz.* neural network-based, modeling of (morpho)phonological phenomena has seen a resurgence, with questions and approaches from past decades being rediscovered, revisited, and revised (Alderete and Tupper, 2017; Kirov and Cotterell, 2018; Prickett, 2021; Nelson et al., 2021). One such question concerns the representations and processes underlying the human acquisition of inflectional morphology. 35 years ago, Rumelhart and McClelland (1986) proposed a feedforward connectionist model of the acquisition of English past tense morphology, kicking off a lively debate (*cf.* (Pinker and Prince, 1988) and much subsequent literature) and setting the stage for a burst of work in connectionist approaches to morphophonology, covering a wide variety of architectures, tasks, and training methods: feed-forward vs recurrent, "meaning" to "sound", input-output string, LM-style training, etc. (Touretzky and Wheeler, 1989; Hare, 1990; Joanisse, 2000, among many others).

This debate was revived by Kirov and Cotterell (2018), who re-examine Pinker & Prince's critique of the Rumelhart & McClelland model in the light of the current state of neural network research in computational linguistics, highlighting the development of new architectures and best-practices with respect to optimization, activation functions, etc. They show that a suitably large sequence-to-sequence network (Sutskever et al., 2014) with RNN encoder and decoder learns to model the acquisition of English past tense near-perfectly, addressing P&P's empirical and in-principle criticisms (but *cf.* critiques by Corkery et al. 2019; McCurdy et al. 2020).

In the present work we are less interested in the empirical successes (in the usual sense of performance on a held-out test set) of such models, but instead focus on the size of the model necessary for this performance and its implications for interpretability. That is, we ask what it is that we learn from the putative success of a large model at a given task; *what does this modeling success tell us about (a) phonology, (b) RNNs, or (c) how RNNs "do" phonology?*

A common approach to NN model interpretation involves extracting state vectors or embeddings from a trained model and performing some form of clustering on them for subsequent visualisation. As the dimensionality of such vectors is often high, a usual first step involves dimensionality reduction via e.g. PCA, UMAP, or some other approach.

In the work described here we investigate the following questions: *what is the smallest model(s) that can reasonably be said to "learn" the task at hand?* and *can we inspect the representations learned by such networks directly, that is, without the lossiness of intermediate dimensionality reduction?* Rather than investigating past tense acquisition directly, we focus on a somewhat more complex and interesting phenomenon, also characterized by morphophonological alternations.

## 2 The domain: vowel harmony

Across many (typologically unrelated) languages and language families one observes co-occurrence restrictions on the distribution of vowels within some phonological domain. When the distributional restrictions are such that the vowels in a language can be grouped into disjoint sets (generally based on pho-

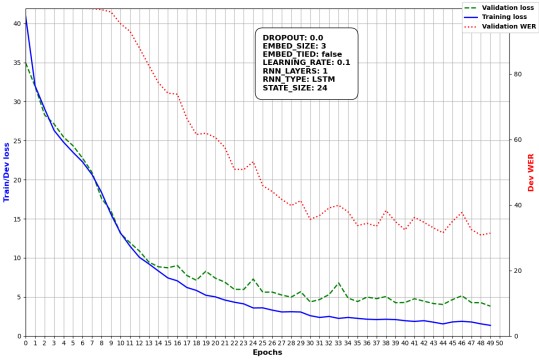

Figure 1: Turkish learning curve with 1000 inputs, 24-d hidden state, and 3-d phone embeddings

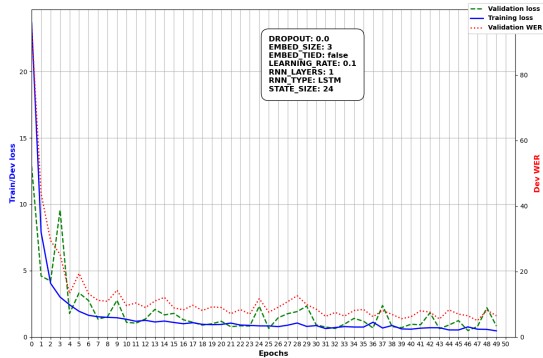

Figure 2: Turkish learning curve with 10000 inputs, 24-d hidden state, and 3-d phone embeddings

netic/phonological features), with words containing vowels from only one of the sets, we use the descriptive label *vowel harmony* (van der Hulst, 2016). Harmony is phonologically interesting because it inherently non-local, skipping intervening consonants. The Turkish example in Table 1 illustrates the basic phenomenon; note that the plural and genitive affixes have variable surface realizations which agree in backness with the stem vowels.

| Lemma | PL | GEN.PL |
|-------|-----|--------|
| savaş | savaşlar | savaşların |
| ipek | ipekler | ipeklerin |

Table 1: Turkish palatal harmony

In some languages with vowel harmony, there are one or more vowels that fail to alternate in context where they would be expected to. These are called *neutral* vowels and are further sub-classified as *opaque* or *transparent* according to whether or not their presence induces further restrictions on the surface forms of co-occurring vowels.

The Finnish examples in Table 2 demonstrates transparent neutrality:

| Surface form | Gloss |
|--------------|-------|
| värttinällänihän | "with my distaff!" |
| palttinallanihan | "with my thin linen!" |

Table 2: Transparency in Finnish vowel harmony

As in the previous Turkish suffixes, the adessive (instrumental) case marker and the emphatic tone particle have two surface realisations — *llä/lla* and *hän/han* — but here the possessive suffix *ni* fails to alternate, and moreover does not trigger any further alternations (cf. the backness of the emphatic in the second example, which on a spreading analysis has "passed through" the intervening front vowel).

## 3 The task

Vowel harmony is generally treated in the linguistic literature as a phonological phenomenon, that is, analyses of and explanations for the observed co-occurrence patterns are stated in terms of phonological theoretical primitives and processes (e.g. feature spreading or constraints enforcing agreement). Here we adopt a *morpho*phonological approach, in concordance with recent iterations of the SIG-MORPHON Shared Task on morphological inflection (Cotterell et al., 2016, et seq.). In these tasks, participants are provided with lemmas and morphological specifications using a broad-coverage tagset (Sylak-Glassman et al., 2015), and must create computational models that learn to output correctly inflected surface forms.

## 4 The data

We use a subset of the data from the SIG-MORPHON 2018 Shared Task (Cotterell et al., 2018) on morphological inflection, specifically the data from Turkish and Finnish. The data files include lemmas along with morphological specifications via UniMorph tags (Sylak-Glassman et al., 2015), and correctly

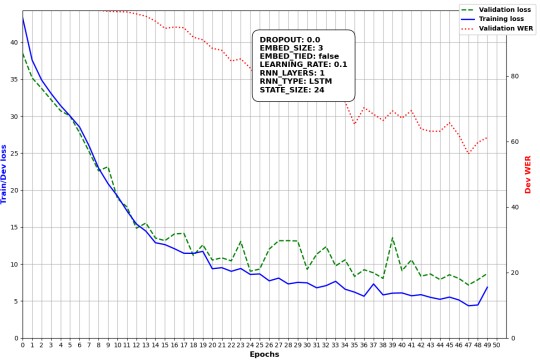

Figure 3: Finnish learning curve with 1000 inputs, 24-d hidden state, and 3-d phone embeddings

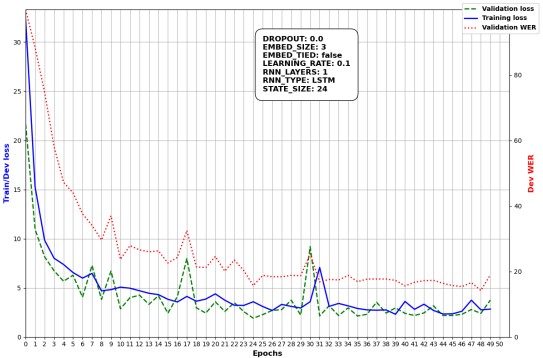

Figure 4: Turkish learning curve with 10000 inputs, 24-d hidden state, and 3-d phone embeddings

inflected surface forms:

| Lemma | Inflected | Morph tags |
|-------|-----------|------------|
| koşucu | koşucularım | N;NOM;PL;PSS1S |
| kart | kartından | N;ABL;SG;PSS2S |

Table 3: Data sample from SIGMORPHON 2018 Shared Task 1

For each language, there are several data files: 3 sets of training data, a development set (for model hyperparameter tuning), and an evaluation set. The 3 training sets contain 100, 1000, and 10000 items, respectively, used to assess model performance under different resource restrictions.

## 5 The model

Kirov and Cotterell (2018) attack the problem of the English past tense with a seq2seq network leveraging RNN-based encoder and decoder with the following specifications:

- 2-layer biLSTM encoder
- 100d LSTM state (e.g. 200d hidden state for fwd and bwd LSTM)
- 300d phone embeddings
- 2-layer biLSTM decoder
- Bahdanau attention (Bahdanau et al., 2016)
- beam search decoding

In the work here we adhere generally to this architecture, but with single-layer RNNs, a unidirectional decoder LSTM, greedy decoding, all parameters scaled down, in some cases significantly (see below for details).

- 1-layer biLSTM encoder
- 1-layer LSTM decoder
- small (see below) phone embeddings
- Bahdanau attention
- greedy decoding

In order to investigate the degree to which our model can be scaled down from K&C's, we opt to target the embedding layer most aggressively, finding that it can be reduced significantly while retaining good performance. In addition, we will see that four to five-fold reductions in LSTM state sizes still result in convergent learning.

## 6 Results

Figures 1, 2, 3, and 4 show learning curves for Turkish and Finnish on the medium and large training sets, respectively, for networks with 24-d LSTM states and 3-d embeddings (a reduction of 2 orders of magnitude from the dimensionality of the embedding space used in K&C). The networks are trained to convergence on training loss with early stopping on diverging development set loss. The ultimate performance metric we are interested in is *word error rate* (WER), essentially whole-word accuracy on output phone sequences.

With 1000 training inputs, the learning curve suggests that some aspects of the inflection task are being learned, but performance at convergence falls short of mastery, even on the training set (this at least indicates that our model is not sufficiently overparameterized to simply memorize the training data). With the full 10000 items of training data, we can see

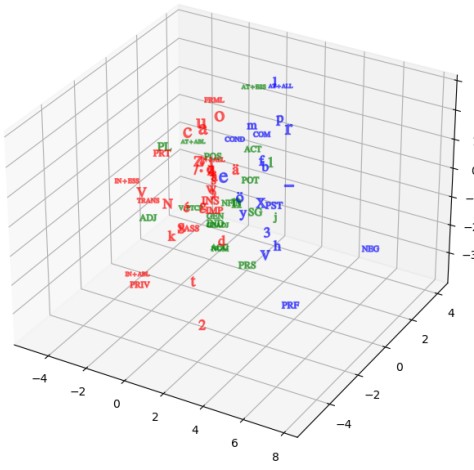
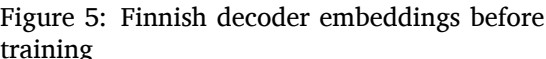
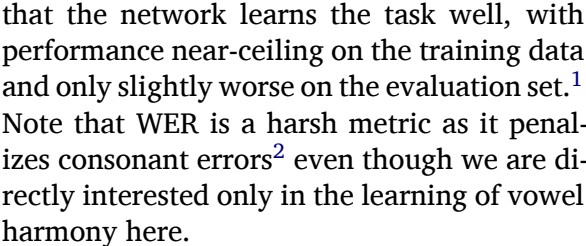

Figure 5: Finnish decoder embeddings before training

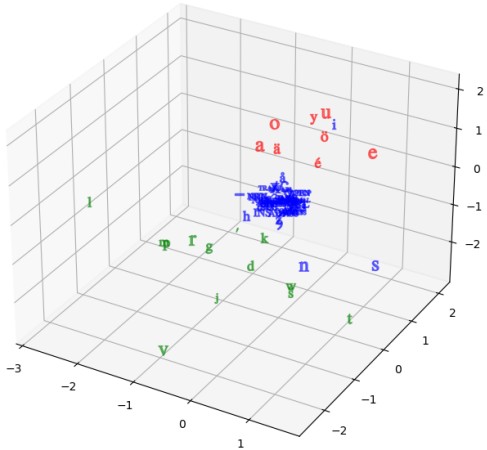

Figure 6: Finnish decoder embeddings after training

that the network learns the task well, with performance near-ceiling on the training data and only slightly worse on the evaluation set.[1] Note that WER is a harsh metric as it penalizes consonant errors[2] even though we are directly interested only in the learning of vowel harmony here.

## 6.1 Learned embeddings

Of perhaps more interest is the structure of the embedding space after learning, which reveals the representations induced from training on this task. Figures 5 and 6 show the distribution of decoder embeddings for the Finnish task before and after training. As expected, the embeddings are initially randomly distributed throughout. After training, we see the emergence of some structure. The three primary clusters correspond to embeddings for the morphological tags, and vowels and the consonants. Within the cluster of vowel embeddings, we can further observe some separation between the front and back vowels, suggesting that distributional information is

enough to learn a latent featural representation. Interestingly, the neutral vowel [i] is clustered with the morphological tags, suggesting that the network is treating is as somehow inert with respect to vowel distribution, but distinct from consonants.

## 7 Discussion

Motivated by recent successes in applying large recurrent neural networks to the task of learning phonological phenomena, we set out to investigate whether a small (orders of magnitude smaller by parameter count) RNN could similarly learn a (morpho)phonologically non-trivial task, and whether the representations learned would be linguistically informative. We demonstrated that such a network can learn vowel harmony, including patterns of transparent neutrality, and moreover that an extremely small (2 orders of magnitude smaller than is typically used) phone embedding space is able to induce linguistically meaningful latent structure; separating vowels, consonants, and morphological tags, and further dividing front and back vowels in accordance with the patterns of harmony investigated. The work here presents many directions for future work: is it possible to go smaller? how well will simi-

---

[1] We ran at least 10 iterations of each network with different hidden seeds; the results here are representative

[2] These errors are rare at convergence but do occur; one network produced the harmonically correct output form `tatbiğinizde` for a target of `tatbikinizde`.

lar networks learn other tasks (e.g. past tense morphophonology as in Kirov and Cotterell 2018)?

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
