# OpenReview forum: "Phonological learning and encoding with small RNNs"
_aclweb.org/ACL/2022/Workshop/CMCL — Submitted to CMCL 2022_

### Official Review · Reviewer_A28a · 2022-03-11
**Promising first exploratory step**

**Rating:** 5
**Confidence:** 4

**Review:**

The authors explore whether small-scale neural networks learn vowel harmony. I am not familiar with the precise literature on modeling phonology using deep learning (so I cannot evaluate the novelty), but the task is straightforward and the phenomenon/task is simple.

I believe this is a nice first exploratory step. For this effort to contribute to the literature, more work needs to be done.

-The authors use a relatively large RNN model originally designed to account for past-tense morphology and scale it down to learn vowel harmony. I don't think we can draw a conclusion from this work as we are comparing apples and oranges.

-The authors show that a model can map some lemmas to their inflections, it's not clear to me what cognitive insight we get from these results? Is the idea to show that humans (babies) learn vowel harmony from distributional information? in which case, does the model reproduce some patterns in children's development?

-There is no quantitative test of vowel harmony, no comparison to previous work?

-The authors select two languages from SIG-MORPHON Shared Task without justifying why these two languages? is it because of their morphological complexity or their typological difference? How does the model behave in the other languages? It is a very simple task and more data is available (about 100 languages in the SIG-MORPHON Shared Task), and I don't see a reason why limit the study to two languages?

-The authors should do a thorough/quantitative investigation of separation between front and back vowels (because that's the latent representation of interest) and not just a qualitative visualization. Also, strangely, the authors do not show results in Turkish to verify if the latent separation exists in Turkish as well.

-No abstract?

---

### Official Review · Reviewer_XYm7 · 2022-03-15
**Nice paper on an understudied phenomenon, but lots of morphophonological assumptions are implicitly built into the model**

**Rating:** 6
**Confidence:** 1

**Review:**

Seems like a nice and well-motivated study, and I appreciate the authors' effort to look into modeling a linguistic phenomenon that is relatively understudied despite being typologically common.

I am a bit unsure what to make of the claims about the model doing well based on a small RNN, because no matter how small you make the model, there is still a lot of morphophonological knowledge implicitly built in to the very assumptions of the model (e.g. Marantz, 2013 [doi: 10.1080/01690965.2013.779385]).

---

### Official Review · Reviewer_8gan · 2022-03-25
**Good paper but I wish there were more explanations about the goal**

**Rating:** 6
**Confidence:** 1

**Review:**

This paper explores the effectiveness of a small RNN when learning a phenomenon of vowel harmony. I guess it's assumed that if a smaller neural network could do the job as well as a larger model, the smaller model is a better choice than the larger one. But still, I wish the authors had explained in more concrete terms what are the benefits of a small RNN. Also, the current phenomenon under investigation is vowel harmony. Would the model results be very different if it is working on a different phenomenon of complex pattern recognition? Is the design of the current model informed by morphophonological conditions of vowel harmony in any way?

---

### Decision · Program_Chairs · 2022-03-29

Reject